# DIVA: Harnessing the Representation Divergence in Unified Multimodal Models for Mutual Reinforcement

**Renjie Lu** [* 1 2] **Xulong Zhang** [* 1] **Xiaoyang Qu** [1] **Shangfei Wang** [2] **Jianzong Wang** [† 1]

## Abstract

Unified Multimodal models (UMMs) built on a single architecture have shown impressive performance in both understanding and generation. We identify a fundamental challenge that lies in inductive biases induced by distinct supervision signals: generation branch prefers high-fidelity, fine-grained representations capable of reconstruction, while the understanding favours semantically discriminative embeddings that remain invariant to task-irrelevant factors. Consequently, optimizing these complementary but non-equivalent objectives within a monolithic backbone leads to mutual impairment instead of enhancement. In this paper, we first analyze the root cause of this interference in unified backbones and reveal a complementary structure in their internal representations. Motivated by the observation, we propose *DIVA*, a self-improved post-training framework that transforms the representation divergence into interior synergy. By explicitly factorizing the visual representation into shared and unique components based on two complementary information flow, DIVA enables both the understanding and generation branches to achieve beneficial transferring while preserving the integrity of unique information from cross-flow interference via mutual information estimation. Despite its generality, our method consistently achieves improvements across visual understanding ($+7.82\%$) and generation ($+8.46\%$). The official code is available at: https://github.com/Jayyy-H/DIVA.

---

*Equal contribution . [1]Ping An Technology (Shenzhen) Co., Ltd., Shenzhen, China [2]University of Science and Technology of China, Hefei, China. Correspondence to: Jianzong Wang <jzwang@188.com>.

*Proceedings of the $43^{rd}$ International Conference on Machine Learning*, Seoul, South Korea. PMLR 306, 2026. Copyright 2026 by the author(s).

## 1. Introduction

Unified Multimodal Models (UMMs) have recently demonstrated impressive capability in both visual understanding

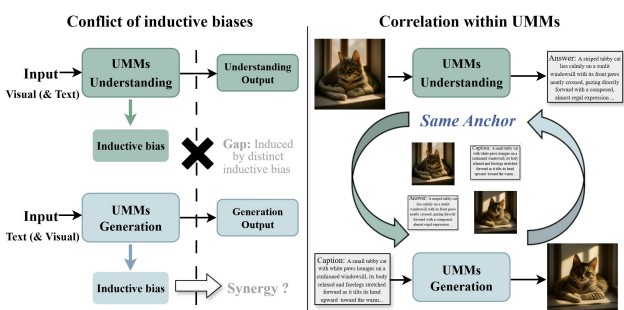

*Figure 1.* Illustration of the gap and base for synergy within UMMs. While the conflict induced by inductive biases from understanding and generation exists, the information flows constructed from same image-text pairs share the semantic anchor, providing the basis for transforming the conflict into mutual reinforcement.

and image generation with a unified architecture (Team, 2024; Pan et al., 2025a; Ge et al., 2024; Wang et al., 2024b; Chen et al., 2025b). While UMMs aim to interleave different tasks within a single backbone and obtain performance improved, existing methods rely on increasingly complex architecture designs, fall short of delivering intrinsic synergy between the capabilities of understanding and generation.

Most existing works (Chen et al., 2025a; Pan et al., 2025b; Chen et al., 2025b) frequently report that optimizing generative objectives negatively degrades the understanding capability. To mitigate this, others (Liao et al., 2025; Qu et al., 2025; Deng et al., 2025) choose to decouple the model component to varying degrees, including separating visual encoders or distinct backbones for different tasks. However, *we argue that such separation compromises the fundamental promise of UMMs*. As indicated by (Gu et al., 2025), an unified architectures and embedding training is essential for integrating the complementary strengths of different branches, enabling the beneficial transfer between understanding and generation. Therefore, the imperative is to resolve the internal conflict within a fully shared architecture to unlock the potential for mutual reinforcement.

As illustrated in Figure 1, although the representation di-

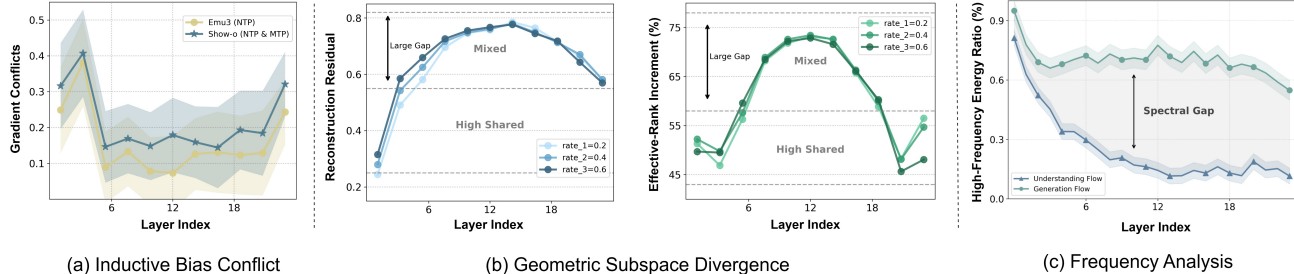

*Figure 2.* Visualization of the representation divergence and synergy. (a) shows the severe conflicts occurs in the shallow and deep layers while the mitigation is observed in the middle layers. Meantime, based on the two information flows that are described in Sec. 3.1, the effective rank between different flows increases in the middle layers and decrease again in the deep layers as presented in (b). And we conduct a frequency analysis in (c) to explore the distinct preferences for information extraction and modeling between understanding and generation branches. The discovery of these phenomena forms the basis of DIVA.

vergence induced by distinct inductive biases often leads to performance degradation, we argue that these different properties actually offer a unique opportunity for conditional mutual reinforcement. The fundamental basis for this synergy is the **shared anchor**: when understanding and generation tasks are constructed from the same data sample, they essentially represent the identical underlying physical reality, despite differing in input-output modalities. The inductive biases can be transformed from conflicts into complementary assets - the semantic-invariance information from the understanding branch provides high-level guidance for faithful synthesis, while the structural sensitivity from the generation branch grounds abstract concepts into fine-grained details. Specifically, we conduct related experiments in Sec. 2 and the results further validated this analysis.

In this paper, motivated by these insights, we propose **DIVA**, a self-improved post-training framework that transforms the conflict between understanding and generation into mutual reinforcement. The core idea is to explicitly factorize the visual representation into shared components that facilitate cross-task transfer, and unique components that preserve task-specific inductive biases. Based on two information flows constructed from understanding and generation branches, we first introduce a collaborative decomposition mechanism. Specifically, we freeze the backbone and training lightweight encoders via factorized logit injection, where the shared encoder learns to transfer semantic skeletons from the counter-flow, and the unique encoder is compelled to capture the remaining flow-specific residuals, constrained by orthogonality. Subsequently, we post-train the UMMs via mutual-information estimation, aligning the shared information while disentangling the unique factors from dual flows across the specific layers. By integrating these constraints with native task supervision, DIVA effectively unlocks the internal synergistic effects within the unified architecture. The main contributions of this paper can be summarized as follows:

- We reveal that the representation divergence induced by inductive biases is not limitation but holds the potential for mutual reinforcement based on same anchor.

- We propose **DIVA** as a self-improved framework, that transforms internal conflict into mutual reinforcement by leveraging controllable transfer between shared and unique information.

- DIVA yields consistent improvements across image understanding, generation and editing, demonstrating its effectiveness and robustness.

## 2. Observation

***Point 1: Task-specfic inductive biases between understanding and generation branch.*** Traditional UMMs are commonly optimized by jointly minimizing an understanding **(Und)** and generation **(Gen)** objectives. Formally:

$$\begin{aligned}\mathcal{L}_{\text{Und}} &= \mathcal{L}(f_\theta(\text{concat}(t_{\text{question}}, h_v)), t_{\text{answer}}) \\ \mathcal{L}_{\text{Gen}} &= \mathcal{L}(f_\theta(\text{concat}(t_{\text{prompt}}, h_v)), I_{\text{gt}}),\end{aligned} \quad (1)$$

where $f_\theta$ is the shared UMM backbone and $h_v$ is the visual embedding extracted by the visual encoder. The textual variables $t_{\text{question}}$, $t_{\text{answer}}$, and $t_{\text{prompt}}$ correspond to the question, response, and generation prompt, respectively, and $I_{\text{gt}}$ denotes the target image. The overall training objective is $\theta^* = \arg\min_\theta(\gamma\mathcal{L}_{\text{Und}} + \lambda\mathcal{L}_{\text{Gen}})$.

The two objectives impose distinct representational preferences, and prior studies(Niu et al., 2025; Pan et al., 2025b) have observed that strengthening one capability (e.g., visual generation fidelity) may degrade the other (e.g., multimodal understanding accuracy), suggesting a persistent form of negative transfer in shared transformers (Team et al., 2025).

***Point 2: Is it possible to transform the conflict into synergy?*** To investigate the internal interactions, we conducted gradient, geometric, and spectral analyses on shared transformers (Wang et al., 2024b; Xie et al., 2024). Gradient

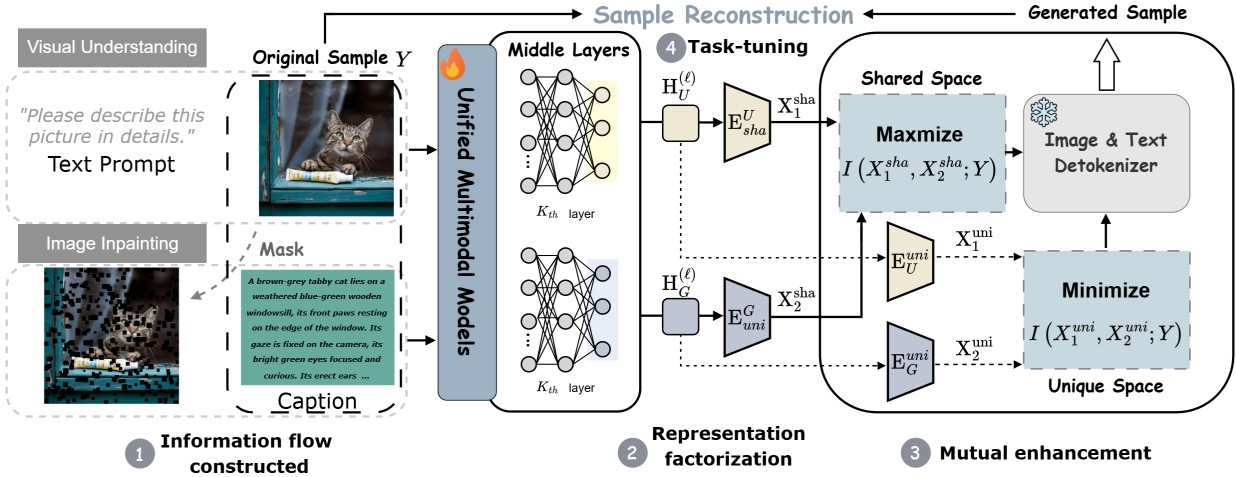

*Figure 3.* **Overview of the self-improved mutual reinforcement (DIVA) pipeline.** We propose a post-training paradigm that explicitly align the shared information, while preserve the integrity of unique information between the understanding and generation flows. Both flows are constructed base on the same sample pair to ensure the shared anchor.

analysis in Figure 2.(a) reveals a inverted parabolic-shaped pattern, that the conflicts are eased in the middle layers while become severe in the shallow and deep layers. To explore the internal interactions, we constructed paired information flows rooted in a common **anchor** (detailed construction of information flows are in Sec. 3.1). Specifically, for a given image-text pair, we extracted layer-wise hidden states from the understanding and generation branches.

Specifically, we employ two geometry-based metrics: Reconstruction Residual and Effective-Rank Increment. The Reconstruction Residual measures the components in the subspace of two information flows that cannot be explained by the information contained in either flow:

$$\mathcal{R}_{\mathrm{res}}(G \mid U) \triangleq \frac{\|G - \Pi_U G\|_F^2}{\|G\|_F^2} \quad (2)$$

where $\Pi_U$ denotes the orthogonal projection operator defined by the PCA basis of information flow. And the Effective-Rank Increment $\Delta\mathrm{ER}$ can be written as:

$$\Delta\mathrm{ER}(H_G \,;\, X \mid H_U) \triangleq \mathrm{ER}(H_{U,G}) - \mathrm{ER}(H_U) \quad (3)$$

As shown in Figure 2.(b), the representations from two flows start with high similarity, significantly diverge into distinct subspaces in the middle layers, and exhibit partial re-coupling in the final layers which is attributed to the shared semantic anchor. This indicates that despite shared weights, the model spontaneously learns to separate task-specific information in the intermediate stages. Frequency analysis Figure 2.(c) further explains this divergence: the understanding flow exhibits a low-frequency bias, rapidly discarding noise to capture global semantics, whereas the generation flow maintains high-frequency energy throughout the depth to preserve fine-grained features.

***Point 3: Analysis and motivation.*** The observations reveal a critical duality: the middle layers spontaneously decouple to accommodate conflicting inductive biases, yet the deep layers re-align, confirming the existence of a ***shared semantic anchor***. This insight drives a fundamental shift: rather than enforcing a monolithic compromise, we propose to explicitly factorize the representations into a *shared space* for the semantic consensus and *unique spaces* for task-specific information. By structuring this decomposition, we can transform the internal interference into a mechanism of controllable mutual reinforcement.

## 3. Methodology

In this section, we present DIVA as a bi-directional self-supervision paradigm for UMMs. Given an image–text pair, first we construct an understanding and generation flows with complementary supervision in Sec. 3.1. Second, we propose a factorized framework that decomposes the dual-view representations into shared and unique components, facilitating bidirectional inductive-bias transfer while mitigating cross-task interference in Sec. 3.2. Finally, we presented DIVA as a two stage post-train method in Sec. 3.3. The overall pipeline is shown in Figure 3.

### 3.1. Preliminary

We first construct two complementary task-induced information flows within the shared transformer backbone $f_\theta$ from the same image-text pair $(I, T)$.

**(1) Understanding Information Flow.** Since the $\mathcal{L}_{\mathrm{Und}}$ aims to project visual features into a language-aligned semantic space, it encourages representations that emphasize global

semantics and structural coherence from the visual information. Therefore, we combine the raw image $I$ with a prompt template $t_{prompt}$ (e.g., "*Please describe this image in detail.*"), and use it as a captioning instruction to elicit a detailed description from the UMM. This captioning supervision induces a low-frequency, global semantic bias in the resulting information flows.

**(2) Generation Information Flow.** To construct this flow, we leverage a self-supervised inpainting task. We apply a random mask $M$ with a ratio $r \in [0.2, 0.6]$ to the original image $I$, yielding a corrupted image $I_{mask} = I \odot (1 - M)$. Using the original paired text $T$ as the semantic condition, the model is asked to reconstruct the missing regions. The $\mathcal{L}_{Gen}$ objective induces a high-frequency, detail-preserving bias in the generation flow.

Motivated by the observation in section 2, we target the middle layers $\mathcal{I}_{mid} = \{l \mid l_{start} \le l \le l_{end}\}$ where task-specific biases are most distinguished. Formally, for any layer $l \in \mathcal{I}_{mid}$, we extract the image-token hidden states $H_U^{img,\ell}, H_G^{img,\ell} \in \mathbb{R}^{N \times d}$ from the understanding and generation flows, respectively. These complementary representations serve as the basis for our mutual improvement paradigm.

### 3.2. Implicit Synergy Via Mutual-Information

**Information Factorization.** Given two task-induced information flows $X_i$ and $X_j$ from the same sample pairs $Y$ derived from the same physical anchor $Y$ (i.e., the image-text pair), we assume that the task-relevant information can be factorized into two types: shared information $\Pi_{sh}$ and unique information $\Pi_{uni}$. The former denotes information that is common across dual flow, while the latter captures information specific to individual flow. Both types of information flow are essential for accurately modeling the unified target $Y$. This factorization can be formalized as follows:

$$I(X_1, X_2; Y) \triangleq \underbrace{\Pi_{sh}}_{\text{Shared Info}} + \underbrace{\Pi_{uni}^i + \Pi_{uni}^j}_{\text{Unique Info}} + \epsilon_{noise} \quad (4)$$

where $\Pi_{uni}^k$ represent the task-relevant information of two information flow, $\epsilon_{noise}$ accounts for irrelevant residuals. This motivates us to align shared factors while preserving unique ones.

As shown in Fig. 3, to compute $\Pi_{sh}$ and $\Pi_{uni}^k$ in Equation 4, we factorize the image-token representations extracted from $\mathcal{I}_{mid}$. For each flow $i \in \{U, G\}$ and layer $\ell \in \mathcal{I}_{mid}$, we first pool image-token hidden states into a layer-wise vector $h_i^{(\ell)} = \text{Pool}\left(H_i^{img,\ell}\right) \in \mathbb{R}^d$, forming a set of layer-wise features $\{h_i^{(l)}, h_i^{(l+1)}, \dots\}$. Then we introduced the shared encoder $E_{sha}^i$ and unique encoder $E_{uni}^i$ which is composed of 3-layer Gated MLPs for each branch, and obtain the shared information $z_{sh}^{\ell,i}$ and unique information

$z_{uni}^{\ell,i}$ as follows:

$$z_{sh}^{\ell,i} = g_{sh}^{(i)}(\ell) \odot \phi_{sh}\left(h_i^{(\ell)}\right), \, g_{sh}^{(i)}(\ell) = \sigma\left(W_{sh}^i \, h_i^{(\ell)}\right),$$
$$z_{uni}^{\ell,i} = g_{uni}^{(i)}(\ell) \odot \phi_{uni}\left(h_i^{(\ell)}\right), \, g_{uni}^{(i)}(\ell) = \sigma\left(W_{uni}^i \, h_i^{(\ell)}\right),$$
$$(5)$$

where $g_{(\cdot)}^{(i)}(\ell)$ is an element-wise soft gate predicted from $h_i^{(\ell)}$, $\phi_{sh}(\cdot)$ and $\phi_{uni}(\cdot)$ are MLPs projections. The training process is presented in Sec. 3.3 which is crucial.

**Mutual Enhancement.** To effectively enable the bidirectional transfer of complementary information between the understanding and generation flows, while preserve the integrity of their unique components, we introduce a mutual-information based learning framework. Let $X_i^s, X_j^s$ denote the shared features produced by Eq. (5) from the two flows, and $X_i^u, X_j^u$ denote the corresponding unique features.

Specifically, we aim to maximize a lower bound on the mutual information between shared representations:

$$I_{sha}(X_i^s; X_j^s) = \mathbb{E}_{\substack{x_i, x_j^+ \sim p(x_i, x_j) \\ x_j^- \sim p(x_j)}} \left[ \log \frac{\exp f(x_i, x_j^+)}{\sum_k \exp f(x_i, x_j^-)} \right],$$
$$(6)$$

where $f(x_i, x_j^+)$ is the optimal critic, and $x_j^+$ refers to the shared features of another information flow from the same sample as $x_i$, while $x_j^-$ denotes the shared features from a different sample.

Maximizing shared information alignment solely is insufficient, as the shared subspace may inadvertently absorb task-specific factors, or the unique subspace may redundantly encode shared semantics, leading to information leakage. To strictly enforce the disentanglement of $\Pi_{uni}$ between $X_i$ and $X_j$, we propose to minimizes the expected upper bound on the unique features $z_{uni}^{\ell,i}$ and $z_{uni}^{\ell,j}$ by utilizing the NCE-CLUB (Liang et al., 2023):

$$I_{uni}(X_i^u; X_j^u) = \mathbb{E}_{x_i, x_j^+ \sim p(x_i, x_j)} \left[ f^*(x_i, x_j^+) \right]$$
$$- \mathbb{E}_{\substack{x_i \sim p(x_i) \\ x_j^- \sim p(x_j)}} \left[ f^*(x_i, x_j^-) \right], \quad (7)$$

where $f^*(x_i, x_j^+)$ is the optimal critic from $I_{NCE}$, used within the $I_{CLUB}$ (Cheng et al., 2020). In practice, we propose an ***asymmetric alignment*** design to stabilize optimization and avoid one-sided dominance of information flow; the exact instantiation is illustrated in Eq. (6) and Eq. (7) together encourage transferable information to concentrate in shared factors while confining view-specific biases to unique factors, enabling the implicit bidirectional synergy under a single backbone. In the following section, we will transition from the theoretical analysis presented above to the practical implementation.

*Table 1.* **Comparison on widely used image understanding and generation benchmarks**. Scores marked with (*) are our reproduced results using 8 random seeds. Models incorporating the DIVA are denoted with **+DIVA**. Detailed scores of GenEval and WISE are provided in Appendix's Sec.C.

| Model | # Params | Types | MMMU | MME | MMBench | MMVet | POPE | GenEval | DPG-Bench | WISE |
|---|---|---|---|---|---|---|---|---|---|---|
| *Understanding Only Models* | | | | | | | | | | |
| LlaVA-v1.5 (Liu et al., 2024a) | 7B | AR | 35.4 | 1488.0 | 78.3 | - | 84.1 | - | - | - |
| Qwen2.5-VL (Bai et al., 2025) | 20B | AR | 58.6 | - | 83.1 | 66.4 | - | - | - | - |
| InstructBLIP (Dai et al., 2023) | 7B | AR | - | 1365.9 | - | 53.2 | 79.4 | - | - | - |
| *Generation Only Models* | | | | | | | | | | |
| SDXL (Podell et al., 2023) | 2.6B | Diff | - | - | - | - | - | 0.55 | 73.75 | 0.43 |
| Qwen-Image (Wu et al., 2025a) | 8B+20B | AR+Diff | - | - | - | - | - | 0.86 | 88.14 | 0.55 |
| SD3-medium (Esser et al., 2024) | 2B | Diff | - | - | - | - | - | 0.74 | 83.81 | 0.42 |
| Infinity (Han et al., 2025) | 8B | VAR | - | - | - | - | - | 0.79 | 86.26 | 0.45 |
| *Unified Multimodal Models* | | | | | | | | | | |
| Janus-Pro* (Chen et al., 2025b) | 7B | AR | 40.6 | - | 69.5 | 49.9 | 86.7 | 0.80 | 84.22 | 0.35 |
| BLIP3-o* (Chen et al., 2025a) | 7B+1.4B | AR+Diff | 56.9 | 1466.2 | 82.5 | 66.3 | - | 0.81 | 80.56 | 0.31 |
| Bagel* (Deng et al., 2025) | 8B+8B | AR+Diff | 54.5 | - | 84.8 | 67.1 | - | 0.84 | 85.04 | 0.52 |
| OmniGen2 (Wu et al., 2025b) | 3B+4B | AR+Diff | 52.6 | 1247.4 | 78.1 | - | 82.4 | 0.80 | 83.59 | 0.36 |
| Emu3 (Wang et al., 2024b) | 8B | AR | 30.7 | 1220.3 | 61.4 | 37.1 | 78.7 | 0.64 | 79.82 | 0.33 |
| Nexus-Gen (Zhang et al., 2025) | 7B | AR | 43.5 | 1279.1 | 70.7 | 45.2 | 83.6 | 0.77 | 81.30 | 0.39 |
| +DIVA | 7B | AR | 49.4 (+5.9) | 1355.3 (+76.2) | 74.9 (+4.2) | 46.6 (+1.4) | 87.4 (+3.8) | 0.83 (+0.06) | 87.87 (+6.57) | 0.45 (+0.06) |
| Show-o* (Xie et al., 2024) | 1.5B | AR | 26.3 | 1097.7 | 48.7 | 32.5 | 73.1 | 0.57 | 69.81 | 0.29 |
| +DIVA | 1.5B | AR | 32.4 (+6.1) | 1206.1 (+108.4) | 51.0 (+2.3) | 33.8 (+1.3) | 79.1 (+6.0) | 0.64 (+0.07) | 76.03 (+6.22) | 0.34 (+0.05) |
| Liquid* (Wu et al., 2026) | 7B | AR | 30.2 | 1321.7 | 57.2 | 36.9 | 77.4 | 0.70 | 80.63 | 0.41 |
| +DIVA | 7B | AR | 34.0 (+3.8) | 1434.9 (+113.2) | 58.9 (+1.7) | 37.8 (+0.9) | 84.5 (+7.1) | 0.81 (+0.11) | 83.47 (+2.84) | 0.44 (+0.03) |

## 3.3. Training Paradigm

In this section, we will transition from the theoretical analysis presented above to the practical implementation of DIVA, a two-stage post-training paradigm. By using native task supervision with cross-task conditioning, we obtain the shared / unique encoders $E_i^s$ and $E_i^u$ in stage 1; Then in Stage 2 we freeze the learned encoders and refines the UMM backbone $f_\theta$ via the proposed asymmetric objectives.

**Stage 1: Task-Driven Encoder Warmup.** We first introduce a *Cross-Task Conditioning* mechanism to exclusively train the $E_s^i$ and $E_u^i$ while freeze the $f_\theta$. The key idea is to inject factorized representations as logit biases: the shared factors provide transferable signals, while the unique factors are encouraged to correct the remaining task-specific residual.

Let $t$ and $v$ index the text-token and image-token positions used in the corresponding losses, $h_\theta(\cdot)$ denotes the logit network of UMMs. For the understanding and generation flows target the same sample, we extract the task-supervised logit blocks by slicing the output logits: $s_U := h_\theta(\cdot)[:, t] \in \mathbb{R}^{V_t \times L}$ and $s_G := h_\theta(\cdot)[:, v] \in \mathbb{R}^{V_v \times M}$, where t and v index the text-token and image-token positions used in the corresponding losses, respectively. Then we obtain the shared factors $z_{\text{sh}}^{\ell,U}$ and $z_{\text{sh}}^{\ell,G}$ via Eq. (5), and inject them as:

$$\tilde{s}_U = s_U + A_U z_{\text{sh}}^{\ell,G} + B_U z_{\text{uni}}^{\ell,U},$$
$$\tilde{s}_G = s_G + A_G z_{\text{sh}}^{\ell,U} + B_G z_{\text{uni}}^{\ell,G}, \quad (8)$$

where $A_U, A_G, B_U, B_G$ are low-rank matrix shared across all layers in $\mathcal{I}_{mid}$ and learned together with the encoders.

We train $E_{uni}$ and $E_{sha}$ by minimizing the native task losses computed on $\tilde{s}_U^{(\ell)}$ and $\tilde{s}_G^{(\ell)}$. Specifically, to prevent the

unique encoder $E_{uni}$ from redundantly encoding shared factors, we add the orthogonality constraints:

$$\mathcal{L}_\perp = \sum_{i \in \{U,G\}} \left\| (\mathbf{z}_{\text{sh}}^i)^\top \mathbf{z}_{\text{uni}}^i \right\|_F^2. \quad (9)$$

In practice, we adopt a simple schedule that warms up the shared-only conditioning before enabling the unique-residual injection. The details about the training process can be seen in Appendix's Sec. A.

**Stage 2: Backbone Fine-Tuning.** After obtain the encoders, we unfreeze the backbone $f_\theta$ and refine it using the mutual-information objectives in Eq. (6) and (7). In practice, directly applying symmetric alignment can be unstable, as the losses of different tasks may differ significantly in scale, leading to one-sided dominance. To avoid this, we adopt an asymmetric alignment with stop-gradient, yielding two directed objectives:

$$\mathcal{L}_{U \to G} = - \log \frac{\exp(\text{sim}(z_{\text{sh}}^U, \text{sg}[z_{\text{sh}}^G])/\tau)}{\sum_j \exp(\text{sim}(z_{\text{sh}}^U, \text{sg}[z_{\text{sh}}^{G,j}])/\tau)},$$
$$\mathcal{L}_{G \to U} = - \log \frac{\exp(\text{sim}(z_{\text{sh}}^G, \text{sg}[z_{\text{sh}}^U])/\tau)}{\sum_j \exp(\text{sim}(z_{\text{sh}}^G, \text{sg}[z_{\text{sh}}^{U,j}])/\tau)}. \quad (10)$$

The stop-gradient operator sg[·] prevents the target view from being updated within each directed term, improving optimization stability under heterogeneous task scales. Overall, we combine the above losses to optimize the UMMs:

$$\mathcal{L}_{total} = \mathcal{L}_{U \to G} + \mathcal{L}_{G \to U} + \mathcal{L}_{uni} + \mathcal{L}_{Und} + \mathcal{L}_{Gen} \quad (11)$$

where $\mathcal{L}_{uni}$ denotes the minimization of upper bound function presented in Eq. 7.

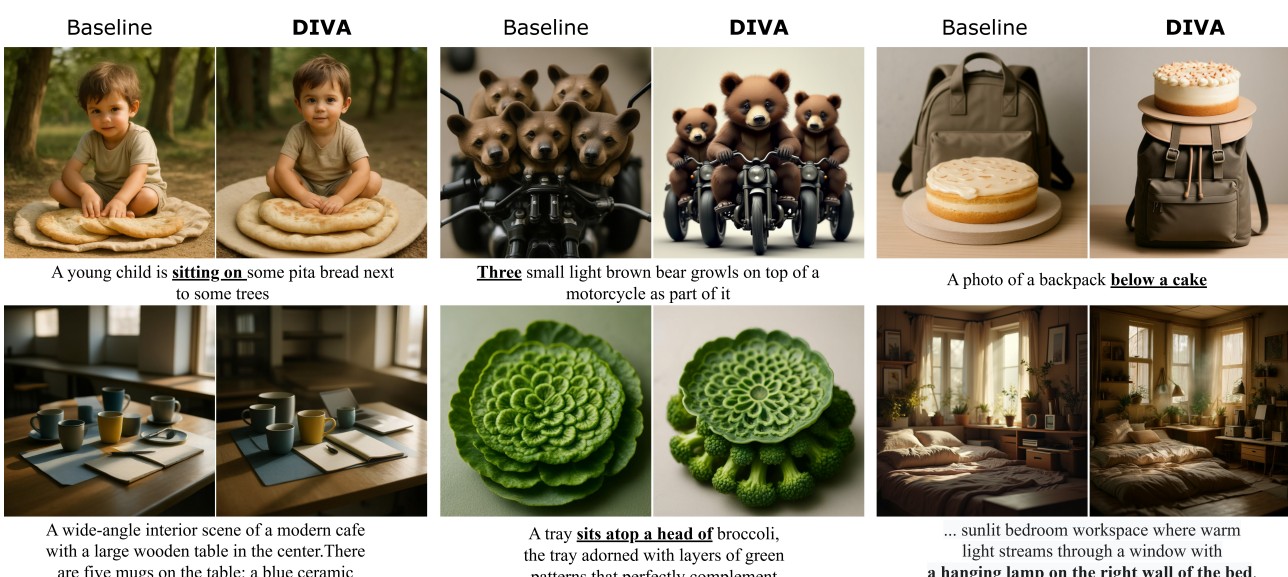

Baseline      **DIVA**      Baseline      **DIVA**      Baseline      **DIVA**

A young child is **sitting on** some pita bread next to some trees

**Three** small light brown bear growls on top of a motorcycle as part of it

A photo of a backpack **below a cake**

A wide-angle interior scene of a modern cafe with a large wooden table in the center.There are five mugs on the table: a blue ceramic mug is to the left of **the closed silver laptop**, a yellow mug is between the blue mug and a folded newspaper.Sunlight enters ...

A tray **sits atop a head of** broccoli, the tray adorned with layers of green patterns that perfectly complement the shape of the broccoli.

... sunlit bedroom workspace where warm light streams through a window with **a hanging lamp on the right wall of the bed**. A neatly arranged desk with small storage boxes, a mirror and some other furniture sits right beside the bed, blending rest ...

*Figure 4.* Qualitative results on image generation. We use Nexus-Gen as baseline for comparsion. It can be observed that after post-train with DIVA, the model's ability of handling the complex attribute, spatial layouts and multiple objectives has significant improved.

*Table 2.* Quantitative comparison for the ablation study about the impact of data quality and method effectiveness, training mechanism of DIVA, sensitivity to middle-layer range, architecture of shared/unique encoders, and mask patterns, where **Bold values** denote the best result within each group.

| Configs | MMMU | POPE | GenEval | DPG-Bench |
|---|---|---|---|---|
| *Data quality and method effectiveness* | | | | |
| Base | 26.3 | 73.1 | 0.69 | 69.81 |
| Base+SFT | 26.8 | 74.5 | 0.67 | 70.75 |
| Base+*DIVA* | **32.4** | **79.1** | **0.75** | **76.03** |
| *Mechanism of DIVA* | | | | |
| w/o $I_{uni}$ | 28.3 | 75.8 | 0.70 | 71.58 |
| w/o sg[·] | 31.7 | 78.2 | 0.73 | 74.92 |
| *Sensitivity to middle-layer range* | | | | |
| Mid-Layer (9–17) | 31.5 | 78.4 | 0.72 | 73.36 |
| Mid-Layer (8–18) | 32.4 | **79.1** | **0.75** | **76.03** |
| Mid-Layer (7–17) | 32.2 | 78.7 | 0.72 | 74.70 |
| Mid-Layer (7–19) | **32.5** | 79.0 | 0.74 | 75.09 |
| *Architecture of shared/unique encoders* | | | | |
| Linear+LN | 29.4 | 75.9 | 0.71 | 72.37 |
| Transformer | 32.1 | **79.2** | 0.74 | 75.65 |
| *Mask patterns* | | | | |
| Contiguous | 24.7 | 69.6 | 0.70 | 68.22 |

*Table 3.* Robustness analysis of DIVA on Show-o under different post-training data sources and scales.

| Training Data | Scale | MMMU | MME | GenEval | DPG-Bench |
|---|---|---|---|---|---|
| JourneyDB | 70K | 31.7 | 1175.8 | 0.62 | 75.71 |
| Mixed-Dataset | 70K | 31.9 | 1193.2 | 0.62 | 75.83 |
| Mixed-Dataset | 200K | **32.4** | **1206.1** | **0.64** | **76.03** |

2024; Wu et al., 2026), which unifies the visual encoder and backbone for both understanding and generation tasks; (2) **Mixture-of-Transformers (MoT)** (Deng et al., 2025), assigning a separate generation-oriented transformer while retaining the original language backbone mainly for understanding; (3) **Hybrid Architecture** (Wu et al., 2025b; Chen et al., 2025b;a), including hybrid encoding (*e.g.*, CLIP or SigLIP for understanding and VAE for generation) or hybrid modeling (*e.g.*, fused autoregressive (AR) and diffusion).

**Implementation Details.** We instantiate DIVA on three representative single-backbone UMMs: Nexus-Gen (Zhang et al., 2025), show-o (Xie et al., 2024) and Liquid (Wu et al., 2026). Detailed hyperparameters and optimization settings are summarized in Appendix's Sec. B.

**Training Data.** Due to the limited availability of paired UMM post-training data, we construct a 200K image-text dataset from both understanding-oriented and generation-oriented sources. Specifically, it contains: (1) 60K quality-filtered samples from CapsFusion-120M (Yu et al., 2024) and Infinity-MM (Li et al., 2025), where we preserve the original image-text pairing and refine the captions with

# 4. Experiments and Results

## 4.1. Experimental Setup

**Baselines.** The selected baselines include: (1) **Shared Architecture** (Wang et al., 2024b; Zhang et al., 2025; Xie et al.,

*Table 4.* Sensitivity analysis of DIVA on Show-o under different weights of the unique-information regularization term.

| Config | POPE | MMMU | DPG-Bench | GenEval |
|---|---|---|---|---|
| Base | 73.1 | 26.3 | 69.81 | 0.69 |
| Base+SFT | 74.5 | 26.8 | 70.75 | 0.67 |
| $\lambda_{uni} = 0.4$ | 78.3 | 31.9 | 74.92 | 0.74 |
| $\lambda_{uni} = 0.6$ | **79.1** | **32.4** | **76.03** | **0.75** |
| $\lambda_{uni} = 0.8$ | 78.7 | 32.2 | 75.50 | **0.75** |

Qwen2.5-VL-32B (Bai et al., 2025); (2) 70K samples from JourneyDB (Sun et al., 2023) with their original text annotations; and (3) 70K samples from MidjourneyV6 (CortexLM, 2024), for which we regenerate image-grounded captions using Qwen2.5-VL-32B. For both information flows, the supervision is constructed from the caption or text prompt associated with the same image-text sample. The understanding flow takes the image with a captioning prompt to elicit semantic descriptions, while the generation flow uses the same associated text as the semantic condition for masked-image reconstruction. This design ensures that the two flows are rooted in the same visual-textual anchor, rather than being optimized with unrelated supervision signals. Further details about the construction of training data are provided in Appendix's Sec. B.

### 4.2. Benchmark Evaluation

**Multimodal Understanding.** We assess visual understanding on five standard benchmarks - MMMU (Yue et al., 2024), MMBench (Liu et al., 2024b), MMVP (Tong et al., 2024), MMVet (Yu et al., 2023) and POPE (Li et al., 2023) - to comprehensively assess the model's capabilities in reasoning, perception, and hallucination robustness. As presented in Table 1, our methods demonstrates consistent improvements over the standard single-backbone baseline across all metrics. Notably, the most significant gains are observed on POPE and MME.

**Text-to-Image Generation.** Following the evaluation protocol of Janus-Pro (Chen et al., 2025b), we evaluate image generation with Geneval (Ghosh et al., 2023) and DPG-Bench (Hu et al., 2024). As shown in Table 1, applying DIVA to Nexus-Gen, Show-o, and Liquid leads to stable improvements on these compositional generation tasks. We further include WISE (Niu et al., 2025), a benchmark built from 1,000 knowledge-puzzle prompts that probe whether generated images reflect implicit factual knowledge. Our strategy conducted on WISE yields consistent gains on Show-o and Nexus-Gen, while Liquid shows smaller improvements. Though DIVA is not designed to enhance the model's ability to learn and master world knowledge, the generation branch can learn to better utilize world knowledge attributd to the enhancements in global information consistency and spa-

tial structure perception. Addition results are provided in Appendix's Sec. A.

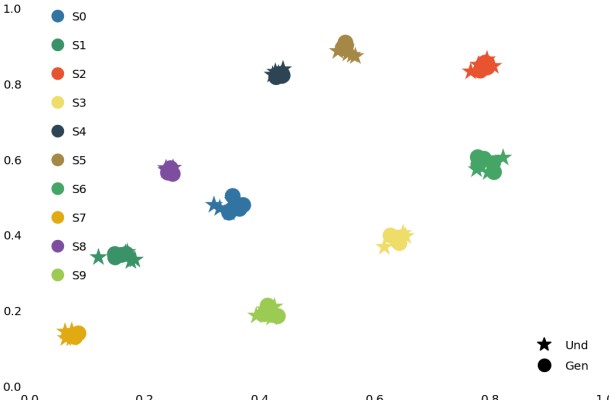

*Figure 5.* The t-SNE visualization of our extracted shared factors. Points of different colors indicate different samples. "Und" implies the shared information from understanding flows and "Gen" represents the shared factors from generation.

**Image Editing.** In addition to Bagel, we conduct experiments on AnyEdit (Yu et al., 2025), UltraEdit (Zhao et al., 2024b) and FLUX.1-Kontext (Labs et al., 2025). As shown in Table 5, *DIVA* conducted on Nexus-Gen consistently outperforms existing baselines across all tasks. It outperforms Bagel, Anyedit and UltraEdit on ImgEdit (Ye et al., 2025), and also obtain improvement on Edit-Bench-EN (Liu et al., 2025). This demonstrates that the improvement of perceiving global information and spatial structure by DIVA can enhance the model's image editing capabilities.

### 4.3. More Results

**Qualitative Results.** Figure 4 demonstrate improvements after conducted on DIVA. The original model often struggles with prompts involving *multiple entities*, *attribute binding*, and *spatial relations*, whereas the DIVA-enhanced model produces images that better follow these constraints. For dense prompts, DIVA more faithfully preserves fine-grained details, reducing the omissions and ambiguous visual bindings observed in the baseline. Additional qualitative results are presented in Appendix's Sec. A.

**Trade-off Analysis.** Figure 6 illustrates the validation losses of understanding and generation tasks under varying weights. We set the weight of understanding loss to 1, and change the weight on the generation loss. The baseline suffers from distinct task conflict, where prioritizing generation performance leads to a significant degradation in understanding. In contrast, the model trained with DIVA consistently obtain lower losses across metrics.

**Visualization of Factorization.** We first randomly select 10 types of test set, each set consists of four sample pairs with

*Table 5.* **Image editing results on ImgEdit and GEdit-Bench-EN benchmarks.** We conducted DIVA on Nexus-Gen to compare with previous methods. The scores of GPT-4o on both benchmarks are reported in (Deng et al., 2025).

| Method | # Params | ImgEdit | | | | | | | | | | GEdit-Bench-EN | | |
| --- | --- | --- | --- | --- | --- | --- | --- | --- | --- | --- | --- | --- | --- | --- |
| | | Rep. | Style | Act. | Ext. | Rem. | Bg. | Add | Comp. | Adj. | Ovr. | SC | PQ | Overall |
| GPT-4o | - | 4.35 | 4.93 | 4.89 | 2.90 | 3.66 | 4.57 | 4.61 | 3.96 | 4.33 | 4.20 | 7.85 | 7.62 | 7.53 |
| AnyEdit | 4B | 2.41 | 2.91 | 2.67 | 1.88 | 2.26 | 2.27 | 3.22 | 1.63 | 2.94 | 2.67 | - | - | - |
| UltraEdit | 4B | 2.86 | 3.81 | 2.98 | 2.16 | 1.43 | 2.84 | 3.48 | 1.93 | 2.81 | 2.99 | - | - | - |
| FLUX.1-kontext | 12B | 4.12 | 4.55 | 4.10 | 1.79 | 2.91 | 3.72 | 3.69 | 2.91 | 3.55 | 3.48 | 6.67 | 7.03 | 6.01 |
| BAGEL | 8B+8B | 3.78 | 4.46 | 4.13 | 1.49 | 3.01 | 3.35 | 3.62 | 2.50 | 3.56 | 3.24 | 7.54 | 6.42 | 6.64 |
| Nexus-Gen | 7B | 3.03 | 3.52 | 2.85 | 2.23 | 1.50 | 3.08 | 3.41 | 1.96 | 2.42 | 2.98 | 5.32 | 4.55 | 4.61 |
| +DIVA | 7B | 3.67 (+0.64) | 3.93 (+0.41) | 3.21 (+0.36) | 2.72 (+0.49) | 1.79 (+0.29) | 3.25 (+0.17) | 3.73 (+0.32) | 2.25 (+0.29) | 2.75 (+0.33) | 3.35 (+0.37) | 5.63 (+0.31) | 4.73 (+0.18) | 4.92 (+0.31) |

similar semantic anchors (distinguishing only from a few attributes). Then we extract the shared factors across all off the understanding flows and generation flows constructed based on these samples. By t-SNE we visualize these factors in Figure 5. The presented results demonstrate our method's ability to obtain the shared information between two flows constructed on the same anchor.

**Empirical Study.** We further compare DIVA with recent post-training methods for UMMs, including RecA (Xie et al., 2025) and UAE (Yan et al., 2025), on both understanding and generation benchmarks. As shown in Table 6, DIVA achieves stronger improvements than RecA under the same SFT-type post-training setting. Specifically, DIVA improves MME by +108.4 and POPE by +6.0, which are substantially larger than the gains obtained by RecA. On the generation side, DIVA also achieves a higher GenEval score, indicating that the proposed factorized mutual-reinforcement objective does not merely enhance visual understanding, but also benefits text-to-image generation. Compared with reconstruction-oriented alignment, DIVA explicitly models the shared and unique information between understanding and generation flows, which enables more balanced improvements across tasks. For UAE, comparable results under the same evaluation setting are not publicly available, so we leave the corresponding entries blank to avoid unfair comparison.

### 4.4. Ablation Study

Considering the computational overhead required for training, we selected Show-o to perform the ablation experiments. The modest scale of this model facilitates a more agile training process, thereby making it feasible to extensively verify the contribution of each module in our proposed method and the results is presented in Table 2.

**Data Quality and Method Effectiveness.** As shown in Table 2, fine-tuning the baseline with our data using standard Supervised Fine-tuning (SFT) brings only marginal changes, suggesting that the post-training dataset itself does not introduce substantial performance gains. In contrast, adding DIVA yields consistent and consistent improvements on both understanding and generation metrics. This gap between Base+SFT and Base+DIVA indicates that the observed gains are largely attributed to our training strategy

rather than data quality or additional fine-tuning alone.

*Table 6.* Comparison with other post-training methods for UMMs. Entries marked with "-" indicate that comparable results under the same evaluation setting are not publicly available, and therefore cannot be fairly reproduced in our setup.

| Method | Types | MME | POPE | GenEval |
| --- | --- | --- | --- | --- |
| RecA | SFT | 1134.8 (+37.1) | 75.7 (+2.6) | 0.63 (+0.06) |
| UAE | SFT+RL | - | - | - |
| DIVA | SFT | 1206.1 (+108.4) | 79.1 (+6.0) | 0.64 (+0.07) |

**Mechanism of DIVA.** To evaluate the importance of key components in our post-training paradigm DIVA, we perform ablations on (i) the unique-information regularization term $I_{uni}$ and (ii) the stop-gradient design used in the shared MI alignment. The results in Table 2 prove the necessity of them. Removing $I_{uni}$ (w/o $I_{uni}$) consistently harms both understanding and generation tasks, indicating that explicitly suppressing cross-flow leakage of unique factors is indispensable for achieving genuine mutual gains. Without this constraint, the optimization is prone to entangle task-specific information and allow shortcut correlations to seep into the shared subspace, which in turn weakens cross-task transfer and leads to broader degradation rather than a single-sided drop. Besides, ablating *stop-gradient* (w/o sg[·]) also yields a noticeable but milder decline, suggesting that its primary role is to stabilize the bi-directional alignment and mitigate gradient interference between the understanding and generation objectives. Together, these ablations support our design rationale: $I_{uni}$ is the key mechanism that enforces a clean shared/unique decomposition to prevent negative transfer, while stop-gradient acts as an important stabilizer that makes mutual-information based sculpting reliably trainable in a unified backbone.

**Sensitivity to Middle-layer Range.** Since DIVA applies the shared/unique factorization and mutual-information objectives on the middle layers where task-specific divergence is most pronounced, we further study its sensitivity to the selected layer range. As shown in Table 2, DIVA performs consistently across a reasonable middle-layer region. The default range of 8–18 achieves the best overall performance, while nearby choices such as 7–17 and 7–19 remain competitive. Meanwhile, indiscriminately enlarging the range does not bring further gains and also increases the training

cost. This verifies that our layer selection is guided by the diagnostic observations in Sec. 2, rather than being a fragile hyperparameter tuned for a single setting.

**Architecture of Shared/Unique Encoders.** The factorization encoders $E_i^s$ and $E_i^u$ play an essential role in early-stage feature mapping. To assess the impact of this design choice, we replace our default Gated-MLP encoders with a standard Linear+LayerNorm mapping and a more heavy Transformer encoder. The results show a clear capacity–stability trade-off. when adopt Linear+LayerNorm as projector shows a notable performance degradation, suggesting that a purely affine mapping with normalization is not expressive enough to capture the non-linear factorization required by the shared/unique information decomposition. In contrast, the Transformer variant performs close to the original solution on most metrics and even achieves a marginal improvements in the POPE benchmark. However, given its substantially higher complexity and optimization burden, this phenomenon indicates that the bottleneck is not simply encoder capacity; Rather, the Gated-MLP already provides sufficient non-linearity to realize effective factorization, while remaining lightweight and stable for post-training. These findings support our architectural choice: a Gated-MLP strikes the right balance between representational power and trainability, making it a practical and effective instantiation of $E_i^s$ and $E_i^u$ for DIVA.

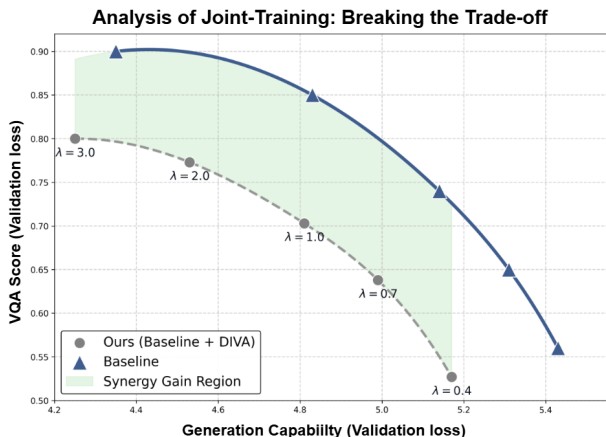

*Figure 6.* Visualization of breakthrough of capability between understanding and generation branches under unified training.

**Mask Patterns.** The results in Table 2 proves that using dispersed random masks to construct the information flow of generation branch is important for mutually enhance. We replace the default random local masking with a same-ratio contiguous block mask and find that performance drops consistently across both understanding and generation tasks. This indicates that block masking weakens the quality and diversity of supervisory signals provided by the generation branch: masking a single continuous region encourages the model to rely more on coarse spatial continuity and local texture propagation, rather than integrating globally distributed semantic and structural features.

### 4.5. Sensitivity and Robustness Analysis

**Robustness to post-training data.** To further disentangle the effect of DIVA from the specific data mixture or captioning pipeline, we additionally evaluate DIVA under different post-training data sources and scales in Table 3. As shown in Table 3, using raw JourneyDB-70K already yields competitive performance, while replacing it with a mixed 70K subset leads to only marginal changes. Increasing the mixed dataset to 200K brings further gains, but the improvement is moderate rather than abrupt. These results are consistent with the Base+SFT versus Base+DIVA comparison in Table 2, suggesting that the observed gains are not mainly attributed to a particular data source or additional fine-tuning alone, but to the proposed factorized mutual-reinforcement training strategy.

**Sensitivity to unique-information regularization.** Although Fig. 6 already shows that DIVA consistently alleviates the conflict frontier between understanding and generation losses, we further conduct an explicit sensitivity study on the weight of the unique-information regularization term. In our main experiments, we keep the original task loss weights of the corresponding base models unchanged, and only use $\lambda_{\text{uni}}$ to control the strength of the NCE-CLUB based unique-information regularization. As shown in Table 4, DIVA consistently outperforms both the base model and the SFT baseline across different values of $\lambda_{\text{uni}}$. The best overall performance is obtained at $\lambda_{\text{uni}} = 0.6$, while nearby settings still maintain clear gains on both understanding and generation benchmarks. These results indicate that DIVA does not rely on a narrowly tuned loss weight to converge or achieve improvements, supporting the robustness of the proposed factorized mutual-reinforcement objective.

## 5. Conclusion and Limitation

DIVA is a self-improved post-training framework designed for achieving synergy in UMMs. It consistently achieves better performance across image understanding, generation, and editing tasks, highlighting the great potential of optimizing UMMs through their internal complementary structures.

**Limitation.** Our current evaluation primarily focuses on models in the 1.5B to 8B parameter range. While we observe consistent gains, validating the scalability of DIVA on larger-scale models remains an important direction to confirm whether our method follows scaling laws. Besides, extending DIVA to broader multimodal settings, such as video and interleaved generation, is worth future exploration.

## Acknowledgements

This work was supported by Shenzhen-Hong Kong Joint Funding Project (Category A) under grant No. SGDX20240115103359001.

## Impact Statement

This work aims to improve unified multimodal models by enabling visual understanding and generation to reinforce each other within a shared backbone. Such models may benefit applications in multimodal assistants, creative content generation, and visual reasoning systems. At the same time, stronger image generation and editing capabilities may also amplify risks such as synthetic-content misuse, biased generation, or visually plausible but incorrect outputs. We therefore encourage responsible deployment with provenance tracking, safety filtering, and careful evaluation under real-world usage scenarios.

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

# A. Related work.

## A.1. Unified Multimodal Models (UMMs)

Vision-Language Models (VLMs) have demonstrated remarkable progress in multimodal understanding and reasoning, enabled by combining Large Language Models (LLMs) with powerful visual encoders (Liu et al., 2023; Team, 2024; Wang et al., 2024a). Motivated by this success, recent research has sought to extend VLMs with image generation capabilities, resulting in the development of **Unified Multimodal Models (UMMs)** (Pan et al., 2025a; Ge et al., 2024; Wang et al., 2024b; Chen et al., 2025b). UMMs aim to combine multimodal understanding and generation within a single backbone, enabling the capacity between understanding and generation interleaved and resulting the improvement of performance across various tasks.

Recent studies can be categorized into the following two types. To combine the inference capabilities of LLMs with the high generative quality of diffusion models, some researches (Ge et al., 2024; Zhou et al., 2024; Zhao et al., 2024a) employ a hybrid strategy of using AR for understanding and diffusion for generation. However, these methods typically introduce additional semantic encoders or complex two-stage designs, sacrificing the uniformity of the architecture and the reciprocal potential of parameter sharing. Furthermore, when performing auto-regressive predictions in continuous embedding spaces, the problem of error accumulation is often encountered, leading to a decrease in generation quality with sequence length.

Other works (Xie et al., 2024; Wang et al., 2024b; Team, 2024; Wu et al., 2026) select discretize visual data into a sequence of tokens, and then jointly model it with text in the same Transformer. Despite its simple architecture, experiments (Zhang et al., 2025; Deng et al., 2025) show that a fully shared Transformer can cause severe gradient conflicts at shallow and deep layers when processing text and images, due to the huge differences in their underlying statistical properties (such as entropy), hindering the effective convergence of the model. Despite the performance are boosted by increasingly complex system designs, the gap between understanding and generation branches within UMMs are the fundamental challenge.

## A.2. Post-Training strategy for UMMs

Supervised fine-tuning with high-quality data (Chen et al., 2025a; Wang et al., 2025) used to be a common and direct practice by utilizing advanced closed-source models (e.g., GPT-4o) to generate large-scale, high-quality image-text pairs. However, this method is limited to its high cost and the risk of distribution shift about generated data. Recently some works choose to explore different techniques to enhance the generation branch of UMMs, given that the understanding branch performs better. For instance, RecA (Xie et al., 2025) leverages reconstruction alignment by conditioning generation on understanding embeddings and using reconstruction losses to bring representations closer. In addition, SRUM (Jin et al., 2025) proposes a fine-grained self-reward framework. Its core lies in utilizing understanding branch as an internal evaluator. By constructing a dual reward system encompassing both global and local dimensions, it guides and optimizes the performance of the generating branch without requiring additional manually labeled data. Compared to these methods, UAE (Yan et al., 2025) introduces a training paradigm based on the auto-encoder perspective: treating the understanding task as an encoder (image to text) and the generation task as a decoder (text to image). By maximizing the fidelity of image reconstruction, it forces the establishment of a bidirectional information flow between understanding and generation, thereby achieving mutual promotion. However, it forcibly uses discrete text as an intermediate information bottleneck, which inevitably leads to the loss of a large amount of pixel-level details. This makes it difficult to perfectly reconstruct the original image by relying solely on text descriptions, thus limiting potential of the model's capabilities.

# B. Implementation Details.

## B.1. Architecture

In Stage 1, we inject factorized representations into task-supervised logit blocks via lightweight readouts. To keep conditioning parameter-efficient, we parameterize each readout matrix as a rank-$r$ factorization:

$$A = PQ^\top, \tag{12}$$

where $P \in \mathbb{R}^{V \times r}$ and $Q \in \mathbb{R}^{d \times r}$, with $V$ denoting the target logit dimension (text vocabulary size $V_t$ or visual-token vocabulary size $V_v$) and $d$ the factor dimension ($d_{\mathrm{sh}}$ or $d_{\mathrm{uni}}$). We use four readouts in total: $A_U, A_G$ for cross-flow shared injection and $B_U, B_G$ for self-flow unique injection,

$$A_U = P_U Q_U^\top, \quad A_G = P_G Q_G^\top, \quad B_U = R_U S_U^\top, \quad B_G = R_G S_G^\top,$$

and share the same readout parameters across all $\ell \in \mathcal{I}_{mid}$ to avoid layer-specific adapters. We set the low-rank dimension to $r = 24$ in all experiments.

**Gated-MLP factorization encoders.** For each flow $i \in \{U, G\}$ and each selected middle layer $\ell \in \mathcal{I}_{mid}$, we first pool the image-token hidden states $H_i^{img,\ell} \in \mathbb{R}^{N \times d}$ into a single vector $h_i^{(\ell)} \in \mathbb{R}^d$. Both the shared encoder and unique encoder adopt a gated-MLP form:

$$z = \mathrm{LN}\Big(g \odot \phi(h)\Big), \qquad g = \sigma(Wh), \tag{13}$$

where $\phi(\cdot)$ is a 3-layer MLPs with a nonlinearity (GELU in our implementation), $W$ is a linear projection producing an element-wise sigmoid gate $g \in (0, 1)^{d'}$, $\odot$ denotes element-wise product, and $\mathrm{LN}(\cdot)$ stabilizes the factor scale.

*Table 7.* Hyperparameters and Settings in the stage 1 of post-training.

| | **Nexus-Gen** | **Show-o** | **Liquid** |
|---|---|---|---|
| **Optimization** | | | |
| Optimizer | AdamW + EMA | AdamW + EMA | AdamW + EMA |
| Learning rate | 2e-4 | 2e-4 | 1.5e-4 |
| LR scheduler | Cosine | Cosine | Cosine |
| EMA decay | (0.99,0.999) | (0.99,0.999) | (0.99,0.999) |
| Weight decay | 0.01 | 0.01 | 0.01 |
| Warmup steps | 300 | 200 | 500 |
| Training steps | 3K | 2K | 5K |
| Grad. accumulation | 5 | 5 | 8 |
| Per-GPU batch size | 6 | 6 | 6 |
| **Trainable modules** | Shared & unique encoders Low-rank readouts | Shared & unique encoders Low-rank readouts | Shared & unique encoders Low-rank readouts |
| **Frozen backbone** $f_\theta$ | ✓ | ✓ | ✓ |
| **Loss weights / schedule** | | | |
| $\lambda_{und}$ | 1.0 | 1.0 | 1.0 |
| $\lambda_{gen}$ | 1.0 | 1.0 | 1.0 |
| $\lambda_\perp$ (Equation 9) | 0.2 | 0.2 | 0.2 |
| Schedule | shared-only $\to$ shared+unique | shared-only $\to$ shared+unique | shared-only $\to$ shared+unique |

## B.2. Training details

Formally DIVA is a two-stage post-training paradigm:

(1) In stage 1 we introduce a cross-task logit biases conditioning mechanism combined with the native task losses to train the shared encoders $E_{sha}^i$ and the unique encoders $E_{uni}^i$. We freeze the backbone and only optimize the factorization encoders and the low-rank logit readouts via the native losses of understanding and generation, together with the orthogonality regularizer. We follow a simple schedule: first by enabling shared-only cross-task conditioning to stabilize the shared encoder $E_{sha}^i$, and then turn on the unique-residual injection so that the unique encoder $E_{uni}^i$ learns to correct the remaining task-specific residuals. The hyperparameters for Stage 1 are summarized in Table 7.

(2) In stage 2 we freeze the factorization encoders and post-train the UMM's backbone using Equation 11, which including the shared alignment (Equation 10) and unique-information regularization (Equation 7) with the native losses of different tasks. As reported in Table 8, we use AdamW with EMA for optimization (with cosine learning-rate schedule), and linearly ramp $\lambda_{sha}$ and $\lambda_{uni}$ from 0 to 0.6 to improve early-stage stability under different objectives.

For Nexus-Gen, we proportionally resize images to approximately $512 \times 512$ resolution for both understanding and generation tasks. Under the DeepSpeed ZeRO-3 framework, the entire post-training process for the 7B model took approximately 74 hours with 8 NVIDIA RTX4090 (24GB) GPUs. For show-o, we proportionally resize images to approximately $256 \times 256$ resolution for both understanding and generation tasks. Under the DeepSpeed ZeRO-2 framework, the entire post-training process for the 8B model took approximately 46 hours with 8 NVIDIA RTX4090 (24GB) GPUs. For Liquid, we

proportionally resize images to approximately $512 \times 512$ resolution for both understanding and generation tasks. Under the DeepSpeed ZeRO-2 framework, the entire post-training process for the 7B model took approximately 89 hours with 8 NVIDIA RTX4090 (24GB) GPUs. Specifically, We use *AdamW* (Loshchilov & Hutter, 2017) for optimization and adopt EMA (Grill et al., 2020) to provide stable targets for bidirectional alignment during post-training.

*Table 8.* Hyperparameters and Settings in the stage 2 of post-training.

| | **Nexus-Gen** | **Show-o** | **Liquid** |
|---|---|---|---|
| **Optimization** | | | |
| Optimizer | AdamW + EMA | AdamW + EMA | AdamW + EMA |
| Learning rate | 5e-5 | 3e-5 | 2e-5 |
| LR scheduler | Cosine | Cosine | Cosine |
| EMA decay | (0.99,0.999) | (0.99,0.999) | (0.99,0.999) |
| Weight decay | 0.01 | 0.01 | 0.01 |
| Warmup steps | 1000 | 800 | 1300 |
| Training steps | 15K | 12K | 20K |
| Grad. accumulation | 10 | 12 | 18 |
| Per-GPU batch size | 6 | 6 | 6 |
| **Trainable modules** | | | |
| Trainable layers | layer 8 - 18 | layer 8 - 18 | layer 9 - 22 |
| **Loss weights** | | | |
| $\lambda_{und}$ | 1.0 | 1.0 | 1.0 |
| $\lambda_{gen}$ | 1.0 | 1.0 | 1.0 |
| $\lambda_{uni}$ | $0 \to 0.6$ | $0 \to 0.6$ | $0 \to 0.6$ |
| $\lambda_{sha}$ | $0 \to 0.6$ | $0 \to 0.6$ | $0 \to 0.6$ |

### B.3. Evaluation details

We briefly introduce the benchmarks we adopted:

**MMMU**: Which is designed to evaluate multimodal models on massive multi-discipline tasks demanding college-level subject knowledge and deliberate reasoning, including four challenges: (1) comprehensiveness: 11.5K college-level problems across six broad disciplines and 30 college subjects; (2) highly heterogeneous image types; (3) interleaved text and images; (4) expert-level perception and reasoning rooted in deep subject knowledge

**MME**: A comprehensive evaluation benchmark for multimodal large language models, measures both perception and cognition abilities on a total of 14 subtasks.

**MMBench**: Contains 2974 multiple-choice questions, covering 20 ability dimensions including: coarse perception, fine-grained single-instance perception, attribute reasoning, relation reasoning and logic reasoning.

**MMVet**: Focuses on the integration of different core vision-language capabilities, including recognition, OCR, knowledge, language generation, spatial awareness, and math.

**POPE**: The POPE benchmark quantifies hallucination rates in object existence verification tasks. It transforms hallucination evaluation into a set of binary classification tasks. Essentially, the MLLMs are posed Yes-or-No questions about the existence of some particular objects in the images, such as "Is there a car in the image?"

**GenEval**: An object-focused framework to evaluate compositional image properties such as object co-occurrence, position, count, and color with 553 prompts.

**DPG**: A specialized evaluation framework for text-to-image models, consisting of 1,065 lengthy and dense prompts that describe multiple objects with complex attributes and relationships. It measures a model's semantic alignment by decomposing these complex instructions into fine-grained evaluation metrics.

**WISE**: The world-knowledge informed T2I evaluation with 1000 structured prompts across 25 subdomains.

*Table 9.* The detailed results in GenEval Benchmark..

| Model | # Params | Single Object | Two Object | Counting | Colors | Position | Color Attribute | Overall |
|---|---|---|---|---|---|---|---|---|
| Janus-Pro | 7B | 0.99 | 0.89 | 0.59 | 0.90 | 0.79 | 0.66 | 0.80 |
| BLIP3-o | 7B+1.4B | 0.99 | 0.91 | 0.62 | 0.87 | 0.84 | 0.65 | 0.81 |
| Bagel | 8B+8B | 1.00 | 0.95 | 0.82 | 0.89 | 0.66 | 0.65 | 0.84 |
| OmniGen2 | 3B+4B | 1.00 | 0.95 | 0.64 | 0.88 | 0.55 | 0.76 | 0.80 |
| Emu3 | 8B | 0.97 | 0.80 | 0.39 | 0.76 | 0.44 | 0.47 | 0.64 |
| Nexus-Gen | 7B | 0.98 | 0.86 | 0.53 | 0.84 | 0.77 | 0.61 | 0.77 |
| +*DIVA* | 7B | 0.98 (+0.00) | 0.95 (+0.09) | 0.60 (+0.07) | 0.89 (+0.05) | 0.84 (+0.07) | 0.70 (+0.09) | 0.83 (+0.06) |
| Show-o | 1.5B | 0.95 | 0.53 | 0.51 | 0.82 | 0.13 | 0.28 | 0.57 |
| +*DIVA* | 1.5B | 0.96 (+0.01) | 0.65 (+0.12) | 0.54 (+0.03) | 0.84 (+0.02) | 0.27 (+0.14) | 0.39 (+0.11) | 0.64 (+0.07) |
| Liquid | 7B | 0.97 | 0.84 | 0.57 | 0.83 | 0.44 | 0.56 | 0.70 |
| +*DIVA* | 7B | 0.98 (+0.01) | 0.91 (+0.07) | 0.66 (+0.09) | 0.91 (+0.08) | 0.71 (+0.27) | 0.70 (+0.14) | 0.82 (+0.11) |

*Table 10.* The detailed results in WISE Benchmark.

| Model | # Params | Cultural | Time | Space | Biology | Physics | Chemistry | Overall |
|---|---|---|---|---|---|---|---|---|
| Janus-Pro | 7B | 0.30 | 0.37 | 0.49 | 0.36 | 0.42 | 0.26 | 0.35 |
| BLIP3-o | 7B+1.4B | 0.33 | 0.34 | 0.31 | 0.27 | 0.28 | 0.20 | 0.31 |
| Bagel | 8B+8B | 0.43 | 0.52 | 0.67 | 0.45 | 0.60 | 0.46 | 0.52 |
| OmniGen2 | 3B+4B | 0.34 | 0.40 | 0.47 | 0.34 | 0.53 | 0.31 | 0.36 |
| Emu3 | 8B | 0.29 | 0.41 | 0.40 | 0.31 | 0.37 | 0.23 | 0.33 |
| Nexus-Gen | 7B | 0.35 | 0.43 | 0.50 | 0.41 | 0.42 | 0.32 | 0.39 |
| +*DIVA* | 7B | 0.35 (+0.00) | 0.47 (+0.04) | 0.64 (+0.14) | 0.46 (+0.05) | 0.53 (+0.11) | 0.34 (+0.02) | 0.45 (+0.06) |
| Show-o | 1.5B | 0.27 | 0.35 | 0.39 | 0.22 | 0.32 | 0.21 | 0.29 |
| +*DIVA* | 1.5B | 0.29 (+0.02) | 0.35 (+0.00) | 0.47 (+0.08) | 0.26 (+0.04) | 0.44 (+0.13) | 0.23 (+0.02) | 0.34 (+0.05) |
| Liquid | 7B | 0.35 | 0.47 | 0.50 | 0.43 | 0.47 | 0.29 | 0.41 |
| +*DIVA* | 7B | 0.35 (+0.00) | 0.45 (-0.02) | 0.60 (+0.10) | 0.44 (+0.01) | 0.53 (+0.06) | 0.32 (+0.03) | 0.44 (+0.03) |

**ImgEdit**: Consists of 1.2 million high-quality image-editing pairs, including 1.1 million single-turn and 110,000 multi-turn samples. The benchmark specifically evaluates models across three dimensions—instruction adherence, editing quality, and detail preservation.

**GEdit-Bench-EN**: Designed to reflect real-world user requirements, covering 11 diverse editing tasks such as background change, subject removal, and text modification. It contains approximately 600 high-quality image-instruction pairs (within a broader dataset scale of 1K−10K samples) and utilizes advanced MLLMs like GPT-4o as automatic evaluators for metrics.

# C. More Experiment Results.

## C.1. The Detailed results on GenEval and WISE

We provide the qualitative analysis in detail across Geneval and WISE benchmark. Table 9 shows DIVA' s consistent performance imporvements across all evaluated aspects. The detailed WISE benchmark results in Table **??** indicates that DIVA primarily enhances the model' s ability to maintain global information consistency while showing modest improvements in reasoning-intensive tasks.

## C.2. Quantitative results

We provide more cases to demonstrate our method' s superiority regarding the performance of generation in Figure 7. Experimental results demonstrate that by guiding the understanding end to maintain global information consistency and possessing strong capabilities in spatial structure layout and complex attribute allocation, the performance of the generation end is enhanced. The concrete text prompts is provided as follows:

(1) A white four-seater sofa.

(2) The bathtub in the bathroom was full of bananas which also existed on the green sofa next to it.

(3) The computer desk space is decorated with mock farm animals on shelves

(4) A lemon-flavored birthday cake.

(5) A cat sits in the foreground of the grass, while other cats walk past behind it.

(6) Two golden dogs lay together on the ground beside the woods.

(7) A red and blue airplane is flying in a field.

(8) Giraffe lying in bed with white pillows.

(9) A red bus. Cartoon style.

(10) A photo of a camera which is angled towards the lens.

(11) A sailboat is trapped in a glass bottle on the ocean.

(12) A clock is placed on the head of a sheep.

(13) Some horses wandered under the Eiffel Tower in Paris.

(14) Two cats standing on snowboards in the Big Ben and London Bridge.

(15) On the table was a white sign with the word "DIVA" written in black lettering.

(16) Two crows standing close to each other. In painting style

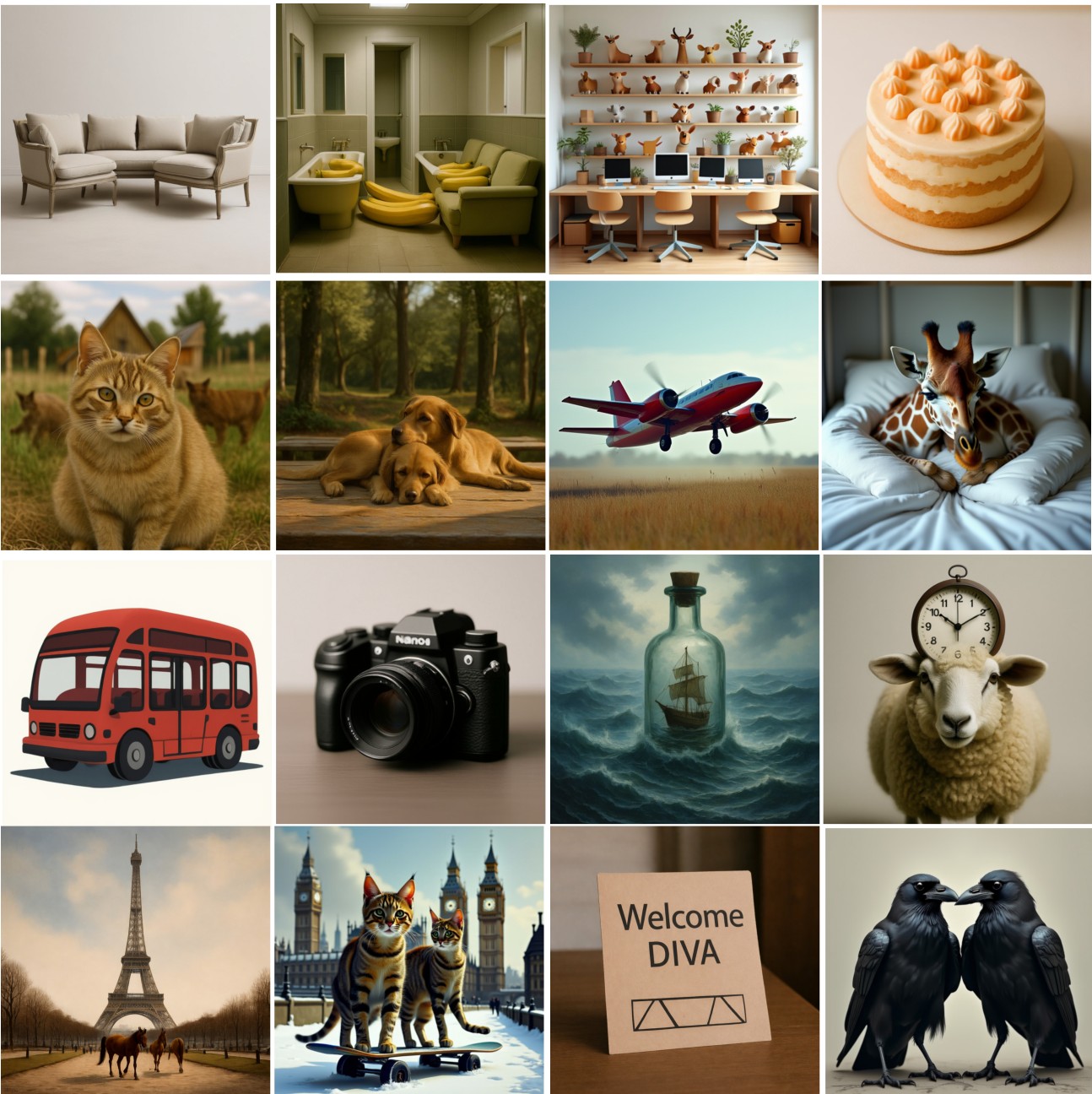

*Figure 7.* Image Generation results. The generating process encompasses multiple dimensions, including world knowledge acquisition, multi-objective scenarios, complex attribute control, spatial layout, and counterfactual generation.

