# OpenReview forum: "DIVA: Harnessing the Representation Divergence in Unified Multimodal Models for Mutual Reinforcement"
_ICML.cc/2026/Conference — ICML 2026 regular_

### Official Review · Reviewer_MMow · 2026-02-21

**Soundness:** 3
**Presentation:** 3
**Significance:** 3
**Originality:** 2
**Overall Recommendation:** 4
**Confidence:** 3

**Summary:**

This paper studies why unified multimodal models (UMMs) with a shared backbone often exhibit mutual impairment between visual understanding tasks and image generation tasks. They observe that the two branches are driven by different inductive biases from the supervision signal, which can cause interface rather than synergy. The authors first analyze layerwise behavior and report a diagnostic pattern: the two branches’ representations diverge in mid layers and partially re-couple in deeper layers. Motivated by their observation, they propose DIVA, a post-training framework that constructs two complementary information flows from a shared image-text anchor. DIVA then factorizes mid-layer visual representations into shared and unique components via gated MLP encoders, aligns shared factors across flows through an InfoNCE-style objective, and suppresses leakage between unique factors using mutual information upper-bound minimization. Empirically, adding DIVA to several recent UMMs yields consistent gains in a suite of multimodal understanding benchmarks and image-generation evaluations, and also improves image editing performance.

**Compliance With Llm Reviewing Policy:**

Affirmed.

**Final Justification:**

My initial concerns have largely been addressed during the rebuttal, and I would keep my initial positive score.

**Key Questions For Authors:**

1. Which components are essential contributions (flow construction, logit-bias warmup, asymmetric stop-grad alignment, unique-factor mutual information upper bound)? What would be the closest existing recipe baseline to contrast against?

2. How sensitive is DIVA to the selected mid-layer range, the choice of the critic for InfoNCE, the number of negatives, and the NCE-CLUB estimator details? Do you observe any instability or collapse modes during training?

3. Can you provide a clearer accounting for the post-training dataset and the exact supervision signals used in each flow (e.g., when using generated grounded captions vs original paired text)? How dependent are gains on the specific teacher model used for caption generation? Do improvements persist if the captioning flow is replaced with a different understanding-oriented supervision?

4. Under matched data and compute, how does DIVA compare to simpler alternatives, such as tuned multi-task loss balancing between understanding and generation losses, or other recent UMM post-training methods beyond the few compared?

**Limitations:**

yes

**Strengths And Weaknesses:**

**Strengths:**

1. The paper is clearly written and easy to follow.

2. The paper targets a real failure mode in shared-backbone UMMs, while explicitly aiming to preserve the benefits of fully shared architectures.

3. The layerwise analysis provides a coherent rationale for the shared-vs-unique factorization rather than forcing a single compromise representation.

4. DIVA is a post-training add-on with lightweight encoders and a two-stage training schedule, and it appears compatible with different UMM families.

**Weaknesses:**

1. The core idea, i.e., shared/private factorization across two views with contrastive alignment and discouraging leakage in private, resembles well-established themes in multi-view and multi-task representation learning. The paper’s novelty is largely in UMM instantiation, but the distinction from related paradigms is not yet fully articulated.


2. It is not fully clear how sensitive the method is to the critic and negative sampling choices when using InfoNCE-style contrastive objective, or whether disentanglement is reliably achieved beyond the shown qualitative t-SNE evidence.


3. The post-training mix combines multiple sources and uses a strong teacher to generate grounded captions. While the SFT vs DIVA ablation helps, more clarity is needed on dataset composition and on whether improvements depend heavily on the teacher or captioning pipeline.


4. The based UMM+DIVA improvements are compelling, but it is hard to separate the method effect from choices such as whether alternative post-training or simpler multi-task balancing would close some of the gap under matched compute and the data used in this work.

---

> ### Author Rebuttal · Authors · 2026-03-31
>
> Thank you for your encouraging feedback and recognition!We have addressed your specific weakness and questions below:
>
> Given the high relevance between your questions and the perceived weakness,we will combine them in our response.
>
> ### [W1]&[Q1]:Core contributions and closest baseline
>
> At the method level,we view the main contribution of DIVA as the **explicit dual-flow formulation** of understanding and generation conflict and **the regulation of shared and unique factors**. In particular, **the upper bound** is used to explicitly suppress cross-flow leakage as we stated in Sec. 4.4.In contrast,**asymmetric alignment** and logit-bias warmup mainly serve as supporting mechanisms that make this framework more trainable and stable.
>
> We also agree that DIVA is related to existing factorization methods in multi-view learning. However,our novelty lies in:factorizes the **heterogeneous** U/G flows in a shared backbone, guided by the **layer-wise diagnosis** in Fig. 2,and further introduces joint shared/unique regularization with asymmetric alignment.Under this view,the closest matched baselines are (1)same-data joint SFT / loss-balanced training(from the method level) (2)Unified-GRPO,RecA(from the perspective of mutual enhancement).
>
> ### [W2]&[Q2]: Sensitivity of key components and training stability
>
> As we stated in Section 3,the selected mid-layer range is **diagnosis-driven** rather than heuristically tuned based on **Figure 2**.To furtherly address your concern,we conducted a sensitive study conducted on Show-o.
>
> |Config|**MMMU**|**POPE**|**GenEval**|**DPG-Bench**|
> |:-:|:-:|:-:|:-:|:-:|
> |Mid-Layer (9-17)|31.5|78.4|0.72|73.36|
> |Mid-Layer (8-18)|32.4|79.1|0.75|76.03|
> |Mid-Layer (7-17)|32.2|78.7|0.72|74.70|
> |Mid-Layer (7-19)|32.5|79.0|0.74|75.09|
>
> The results shows that **DIVA works consistently across a reasonable middle-layer region,while indiscriminately enlarging the span does not yield further gains and increases training cost**.
>
> For the InfoNCE term, we use a **simple similarity-based critic**, and the NCE-CLUB upper bound reuses the same critic instead of introducing an additional learned module,which **keeps the estimator lightweight and limits extra sensitivity**.We also use **a standard in-batch negative setup** rather than a specialized hard-negative design.
>
> Regarding the instability,the main failure mode we identified is **one-sided dominance** under direct cross-flow alignment,which is exactly why DIVA **adopts asymmetric stop-gradient design**.
>
> ### [W3]&[Q3]: Post-training data, supervision signals, and dependence on teacher model for caption generation.
>
> Our post-training dataset contains:(1)60k quality-filtered samples from CapsFusion-120M and Infinity-MM,where we only refine the captions based on the original paired data using Qwen2.5-VL-32B (2)70k samples **directly** from JourneyDB (3)70k samples from MidjourneyV6,where the captions are regenerated by Qwen2.5-VL-32B.For both flows,the supervision is unified as the caption or text prompt associated with the same image-text sample.
>
> We further post-train show-o-1.5B on raw JourneyDB-70k to evaluate the data robustness of DIVA
>
> |Training Data|Data Scale|MMMU|MME|GenEval|DPG|
> |:-:|:-:|:-:|:-:|:-:|:-:|
> |JourneyDB|70k|31.7|1175.8|0.62|75.71|
> |Mixed-Dataset|70k|31.9|1193.2|0.62|75.83|
> |Mixed-Dataset|200k|32.4|1206.1|0.64|76.03|
>
> The results are aligned with what we claimed in Sec 4.4 that the observed gains primarily come from our training strategy rather than data quality or additional fine-tuning alone.
>
> We did make a preliminary attempt to construct the understanding flow using VQA-style supervision, but its compatibility with the generation flow was weaker than that of caption-based supervision. We attribute this to the cleaner shared semantic anchor and lower textual entropy of captions, whereas VQA introduces prompt-dependent perturbations that weaken U/G alignment.
>
> ### [W4]&[Q4]: Compare to other UMM post-training methods
>
> Under matched data and compute,we already compare against **same-data SFT**(Table 2) and **same-data RecA**(Table 4) which are accessible.For more post-train methods,most of them are unavailable since **there are not open-source code as well as the API provided** except RecA. For a intuitive comparison, we provided the overall performance based on the data reported in the corresponding papers **in our reply to Reviewer AQqN in W1**.
>
> For simpler alternatives such as tuned multi-task loss balancing, our trade-off analysis in Figure 6 already probes this types of methods by sweeping the relative weight. It shows that standard joint training remains on a clear conflict frontier, while DIVA consistently lowers both sides. The reason is that loss balancing mainly acts at the gradient level, whereas DIVA addresses the conflict at the representation level.
>
> ***Hope our explanation and experiments can address your inquiries. We will integrate all your valuable comments into our revision!***

---

> > ### Author Rebuttal · Reviewer_MMow · 2026-04-02
> >
> > Thank you for your response. My concerns have largely been addressed, and I would keep my positive score.

---

> > > ### Author Response · Authors · 2026-04-03
> > >
> > > Thanks for your thoughtful and constructive feedback again! We are very encouraged that our rebuttal has addressed your concerns, and especially appreciate your recognition of DIVA as a post-training framework that tackles the internal conflict in native unified architectures while preserving the benefits of a fully shared backbone.We will incorporate these clarifications and analyses above carefully into the revised manuscript. If you feel that these additions further strengthen the paper, we would sincerely appreciate it if you would consider raising the score.

---

### Official Review · Reviewer_AQqN · 2026-03-11

**Soundness:** 3
**Presentation:** 3
**Significance:** 3
**Originality:** 3
**Overall Recommendation:** 4
**Confidence:** 4

**Summary:**

This paper introduces DIVA, a post-training method aiming to address the representation conflict between understanding and generation tasks in unified multimodal models. By employing mutual information estimation, DIVA achieves substantial improvements on baselines in both tasks.

**Compliance With Llm Reviewing Policy:**

Affirmed.

**Final Justification:**

The rebuttal has resolved my concerns.

**Key Questions For Authors:**

1. What is the training cost of DIVA compared to other UMM's post-training method, such as RecA[1] and other methods?
2. Can the author provide detailed comparisons of these instead of the baseline UMM models?

[1] Xie, Ji et al. “Reconstruction Alignment Improves Unified Multimodal Models.” ArXiv abs/2509.07295 (2025): n. pag.

**Limitations:**

yes

**Strengths And Weaknesses:**

Strength:
1. This paper is well-motivated, and the proposed method is quite novel. This inductive bias conflict is common in umms, and leveraging its own characteristics to solve its own problems is inspiring.

2. The experimental results are thorough and convincing.  The author evaluates across understanding, generation, and editing tasks. DIVA demonstrates its effectiveness across multiple backbones. The ablations on removing the unique information term and stop-gradient are directionally useful.

3. The writing of this paper is clear and easy to understand. The shared/unique factorization, combined with cross-flow transfer and unique-factor suppression, is conceptually reasonable.


Weakness:
1. The experimental comparisons should be against strong post-training or improvement methods applied to the same backbones under matched data and compute budgets. Current comparisons seem not to be fair.
2. Typos in the current manuscript: “We identify a fundamental challenge lies in...”, “consistently improvements”

---

> ### Author Rebuttal · Authors · 2026-03-31
>
> Thank you for your valuable suggestions! We are grateful for your positive assessment of our work, particularly your appreciation of DIVA's novelty and strong empirical performance as a plug-and-play component for future UMM pipelines. We have incorporated your suggestions into our revision and provide detailed responses to your concerns below:
>
>
> ### [W1]:Comparisons against strong post-training methods
>
> We thank the reviewer for this important comment. We agree that comparisons against strong post-training methods under matched data and compute budgets are the fairest way to assess DIVA.In fact, within the set of runnable and controlled baselines in our setting, we already compare against **same-data SFT (Table 2)** and **same-data RecA** (Table 4). For broader recent UMM post-training methods, we were willing to include more comparisons with them. However, most of them ( including **(1) UAE/Unified-GRPO（2) SRUM（3) UniCorn  (4) UniRL (5) UniMRG**, etc)  are **neither open source nor provide usable APIs**, making it impossible for us to reproduce them and conduct fair comparisons under the same experimental configuration, except RecA.
>
> To still facilitate a broader assessment, we additionally summarize **their reported data results** corresponding to the original papers as followed.
>
>
> | Methods | Base model | MMMU | MMBench | GenEval | DPG-Bench |
> | :---: | :---: | :---: | :---: | :---: | :---: |
> | SRUM[1] | Bagel | -0.002% | -0.002% | +1.2% | - |
> | Unicorn[2] | Bagel | +1.89% | - | +5.12% | +3.3% |
> | UniMRG[3] | Show-o | - | +0.5% | +6.06% | +3.28% |
> | UniMRG[3] | Harmon | - | +3.56% | +19.46% | +5.89% |
> | UniMRG[3] | OpenUni | - | +0.3% | +11.23% | +2.98% |
> | UniRL[4] | show-o | +11.2% | - | +18.33% | - |
> | UniRL[4] | Janus | +4.76% | - | +8.33% | - |
> | Unified-GRPO[5] | UniWorld | -0.4% | - | +5% | +5.2% |
> | Unified-GRPO[5] | Janus-Pro | +0.6% | - | +4.3% | +4.7% |
> | **DIVA** | Nexus-Gen | +12.56% | +5.94% | +7.79% | +8.08% |
>
> **It is important to note that we have not been able to actually implement these methods under the same data and computing configuration, except RecA. The experimental comparison settings in the aforementioned papers also support our claim.**
>
> References
>
>  [1] Jin W, Niu Y, Liao J, et al. Srum: Fine-grained self-rewarding for unified multimodal models. arXiv preprint arXiv:2510.12784, 2025.
>
>  [2] Han R, Fang Z, Sun X Y, et al. UniCorn: Towards Self-Improving Unified Multimodal Models through Self-Generated Supervision. arXiv preprint arXiv:2601.03193, 2026.
>
>  [3] Su Z, Wei H, Cen K, et al. Generation Enhances Understanding in Unified Multimodal Models via Multi-Representation Generation. arXiv preprint arXiv:2601.21406, 2026.
>
>  [4] Mao W, Yang Z, Shou M Z. Unirl: Self-improving unified multimodal models via supervised and reinforcement learning. arXiv preprint arXiv:2505.23380, 2025.
>
>  [5] Yan Z, Lin K, Li Z, et al. Unified Multimodal Model as Auto-Encoder. arXiv preprint arXiv:2509.09666, 2025.
>
>
> ### [W2]: Typos error
>
> We appreciate you pointing out the typos, which we will correct in the revised paper !
>
>
> ### [Q1]: Training cost of DIVA compared to other UMM's post-training method, such as RecA
>
> As we stated in our response in W2,since there is no open-source code or API regarding other UMM's post-training method, we can only compare with RecA under the same data and configuration: For show-o, RecA takes about 32 hours with 8 NVIDIA RTX4090 (24GB) GPUs under the DeepSpeed ZeRO-2 framework, while DIVA takes about 46 hours under the same data and configuration. We acknowledge that DIVA requires an additional computational cost compared to RecA.This extended training time is a direct and expected consequence of DIVA's structural design. However, we argue that this moderate increase in training cost is **a highly worthwhile trade-off**. On the one hand, DIVA significantly improves the model's performance on **both the generation and understanding sides** and **outperforms** it on the vast majority of benchmarks(Table 4). On the other hand, it does not incur extra latency during inference. And we will add this detailed efficiency comparison to Appendix in the revised manuscript to ensure full transparency.
>
>
> ### [Q2]:Detailed comparisons of post-training methods
>
>
> Please see our reply to Weakness 1.
>
> ***Hope our explanation and experiments can address your inquiries. We will integrate all your valuable comments into our revision!***

---

> > ### Author Rebuttal · Reviewer_AQqN · 2026-04-03
> >
> > Thank you for your response. My concerns have largely been addressed, and I keep my positive score. Meanwhile, I strongly encourage the authors to include these further analysis in the revised manuscript.

---

> > > ### Author Response · Authors · 2026-04-03
> > >
> > > Thank you for your careful and constructive feedback! We are pleased that our response has addressed your concerns. We particularly appreciate your recognition of the our paper’s motivation, the thorough evaluation and the factorization design. Your suggestions on training cost and more detailed comparisons with existing post-training methods have been especially helpful, and we will make sure to include these analyses and clarifications in the revised manuscript. If you feel that these additions further strengthen the paper, any increase in the score would be very encouraging to us !

---

### Official Review · Reviewer_kVNh · 2026-03-12

**Soundness:** 2
**Presentation:** 3
**Significance:** 2
**Originality:** 2
**Overall Recommendation:** 4
**Confidence:** 4

**Summary:**

This paper proposes DIVA, an adaptive post-training framework intended to mitigate representation divergence between understanding and generation tasks in Unified Multimodal Models (UMMs) by explicitly decoupling visual features into shared and task-specific components optimized via mutual information objectives.

**Compliance With Llm Reviewing Policy:**

Affirmed.

**Final Justification:**

All my concerns have been adequately addressed in the rebuttal; therefore, I have raised my final evaluation score accordingly.

**Key Questions For Authors:**

See weakness

**Limitations:**

yes

**Strengths And Weaknesses:**

Strengths:

1.The explicit factorization of visual representations into shared and task-specific components provides a conceptually reasonable lens through which to view the ubiquitous modality and task conflicts inherent in UMMs.

2.The two-stage post-training strategy, specifically the introduction of asymmetric alignment and a stop-gradient mechanism, offers a practical engineering solution to improve optimization stability when jointly training on highly heterogeneous tasks.

Weaknesses:

1.The paper focuses primarily on reporting performance gains of existing models after applying DIVA, but lacks a comprehensive comparison with other state-of-the-art post-training methods, limiting the ability to establish the method's relative superiority.

2.The paper attempts to compare DIVA with other post-training methods in Table 4, but inexplicably omits all performance metrics for the "UAE" baseline (leaving them blank), rendering its inclusion meaningless and severely undermining the completeness and academic rigor of the empirical evaluation (further evidenced by glaring typos such as "Comparion" in the caption).

3.The paper lacks rigorous quantitative or visual validation of the proposed factorization. While the method claims to disentangle shared and unique representations, no direct evidence is provided—such as t-SNE visualizations of shared features extracted across different information flows, or quantitative measures of orthogonality between unique and shared components. The absence of such analyses severely undermines the credibility of the core assumption: that clean disentanglement enables mutual reinforcement rather than mere compromise. Without these validations, the claimed synergistic improvement remains speculative and insufficiently supported.

4.The ablation study is conducted exclusively on the 1.5B Show-o model, which significantly deviates from the current mainstream evaluation practice on 7B–cale unified multimodal models (e.g., Nexus-Gen, Liquid). This limits the generalizability of the findings and raises doubts about whether the proposed framework scales effectively to larger architectures.

5.The comparison is limited to standard SFT, which only shows that DIVA mitigates negative transfer but does not provide evidence for the claimed mutual reinforcement between understanding and generation. Without demonstrating that improvements in one task causally benefit the other (e.g., via cross-task correlation or joint trajectory analysis), the core mechanism remains unverified.

6.The paper omits sensitivity analysis of  loss components. Without such analysis, it remains unclear whether DIVA requires extremely fine-tuned settings to converge or achieve gains, casting doubt on its stability, robustness.

---

> ### Author Rebuttal · Authors · 2026-03-31
>
> Thank you for your feedback.Your comments have been highly beneficial in addressing the core issues of our manuscript.In response, we have conducted supplementary experiments and wish to further clarify the related points.
>
> ### [W1]:Comparison with other post-training methods.
>
> Please see our reply to Reviewer AQqN in Weakness 1.
>
> ### [W2]:The omission of performance metrics of UAE and typos error.
>
> We thanks for pointing this out and agree that leaving the UAE row blank may create confusion. Our original intention was only to include UAE as a **method-type reference**,since we could not implement it under the same compute settings due to **the lack of public code or API**. For addressing your concern, we will use the reported data in the original paper to fill in the blanks in Table 4, and strictly indicate the data sources.
>
> We also appreciate you for point out the typo errors,which we will correct in the revised manuscript!
>
> ### [W3]:Correction of missing t-sne experiments and orthogonality assessment.
>
> We respectfully clarify that **the requested analysis of t-SNE visualization of shared features across different information flows, is already provided in Figure 5 of current paper, and Section 4.3 has explicitly described the procedure**.
>
> We have to clarify that we do not overlook the evaluation of orthogonality.(1)In Stage 1, DIVA already imposes a direct orthogonality constraint between shared and unique factors via Eq. (9), which directly penalizes the correlation between $z_{sh}$  and $z_{uni}$. (2) Figure 5 shows that the extracted shared factors from understanding and generation flows **cluster around the same anchor**, indicating that it can effectively separate unique factors (3)Ablation study in Table 4 furtherly demonstrates that w/o $I_{uni}$ causes consistent degradation on both sides, which is exactly the failure mode expected when leakage increases.
>
> ### [W4]: Ablation study on a larger-scale model
>
> We agree that the systematic ablations were mainly conducted on show-o-1.5B for efficiency in the original paper. To directly address your concern about scalability, we further add **a 7B-scale ablation on Liquid**:
>
> |Configs|MMMU|POPE|GenEval|DPG-Bench|
> |:-:|:-:|:-:|:-:|:-:|
> |Base|30.2|77.4|0.70|80.63|
> |Base+**DIVA**|**34.0**|**84.5**|**0.81**|**83.47**|
> |**Mechanism of DIVA**|||||
> |w/o $I_{uni}$|31.1|79.4|0.74|81.16|
> |w/o sg[$\cdot$]|33.0|82.6|0.78|82.44|
> |**Architecture of shared/unique encoders**|||||
> |Linear+LN|31.7|80.2|0.74|81.39|
> |Transformer|33.6|83.4|0.79|82.26|
> |**Mask patterns**|||||
> |Contiguous|27.1|73.8|0.71|80.84|
>
> The results remain consistent with those on Show-o-1.5B (in Table 2), strengthening the generalizability of our findings.
>
> ### [W5]:Limited comparison.
>
> We respectfully clarify that Figure 6 already provides evidence beyond merely mitigating negative transfer:SFT remains on **a clear conflict frontier**, whereas DIVA consistently achieves lower validation losses on both sides, which indicates **a reshaped U/G trade-off** rather than simply choosing a different compromise point.
>
> We have to clarify that our claim is mutual improvement within a unified model,not a formal causal claim that “improving task A directly causes improvement in task B”. Specifically, DIVA enables the model to **use its native capability on one side to benefit the other side without relying on external models or tools**, by controlable information transfer.Moreover, our **ablation study** in Table 2 on $I_{uni}$ also proves that DIVA is not a mere one-sided compromise.
>
> ### [W6]:Omission of sensitivity analysis of loss weights.
>
> We agree that explicit sensitivity assessment makes the robustness claim clearer.We already provides partial sensitivity evidence through the trade-off analysis in Figure 6, where we keep $\lambda_{und}$ fixed and sweep $\lambda_{gen}$: the baseline remains on **a clear conflict frontier**, while DIVA consistently obtains lower validation losses.
>
> Additionally, for all reported DIVA results on different base model,we **retain the original task loss weights($\lambda_{und}$ and $\lambda_{gen}$) consistent** with the weights set during joint training of the original model (as in Table 6), rather than retuning them for each model.
>
> To directly address the reviewer’s concern on loss-component sensitivity, we further add **a sensitivity study of $λ_{uni}$** on Show-o-1.5B.
>
> |Config|**MMMU**|**POPE**|**GenEval**|**DPG-Bench**|
> |:-:|:-:|:-:|:-:|:-:|
> |Base|26.3|73.1|0.69|69.81|
> |Base+SFT|26.8|74.5|0.67|70.75|
> |$\lambda_{uni}=0.4$|31.9|78.3|0.74|74.92|
> |$\lambda_{uni}=0.6$|32.4|79.1|0.75|76.03|
> |$\lambda_{uni}=0.8$|32.2|78.7|0.75|75.50|
>
> These results shows that DIVA does not depend on narrowly tuned loss weights to converge or achieve gains, supporting its robustness.
>
> ***Hope our explanation and experiments can address your inquiries. We will integrate all your valuable comments into our revision!***

---

> > ### Author Rebuttal · Reviewer_kVNh · 2026-04-04
> >
> > All my concerns have been adequately addressed in the rebuttal; therefore, I have raised my final evaluation score accordingly.

---

> > > ### Author Response · Authors · 2026-04-04
> > >
> > > Thank you for reconsidering the paper! We sincerely appreciate your constructive feedback throughout the rebuttal process. Your comments were very helpful in prompting us to clarify the factorization evidence, strengthen several analyses, and enhance the presentation of our main claims. We will carefully incorporate these improvements into the revised manuscript. Thank you again for your time and effort.

---

### Decision · Program_Chairs · 2026-04-30

**Decision:**

Accept (regular)

**Comment:**

The submission addresses the informational divergence across modalities and the challenges it poses for multimodal learning frameworks, where multiple modalities are expected to blend synergistically. The submission starts with an analysis and is followed by proposing a post-training representation factorizing solution. All reviewers are in support of the submission. While they stopped short of finding it transformational, with a level of novelty and applicability that would likely lead to a change in common practices, it had enough merit to warrant exposure at ICML. @the authors, please carefully take into account all reviewers' comments and exchanges, and address them in the camera-ready for greater impact by the paper.